# Robust Instant Angle Speed Measurement for Internal Combustion Engines—A Novel Sensing Suite and Methodology

**DOI:** 10.3390/s22030754

**Published:** 2022-01-19

**Authors:** Ioan Porumb, Romeo Marian, Kutluyil Doğançay, Javaan Singh Chahl

**Affiliations:** UNISA STEM, Australian Research Centre for Interactive and Virtual Environments, University of South Australia, University Boulevard, Mawson Lakes, SA 5095, Australia; romeo.marian@unisa.edu.au (R.M.); kutluyil.dogancay@unisa.edu.au (K.D.); Javaan.Chahl@unisa.edu.au (J.S.C.)

**Keywords:** rotary machinery, small aero piston engine, internal combustion engine, engine-cycle, fine granularity measurement

## Abstract

This paper presents the development and implementation of a novel robust sensing and measurement system that achieves fine granularity and permits new insights into operation of rotational machinery. Instant angle speed measurements offer a wealth of useful information for complex machines in which the motion is the result of multidimensional, internal, and external interactions. The implementation of the proposed system was conducted on an internal combustion engine. The internal combustion engine crankshaft’s angular velocity is the result of the integration of all variables of motor and resisting forces. The crankshaft angular velocity variation also reflects the interaction between the internal thermodynamic cycle of the engine and the plant it powers. To minimise the number of variables, we used for our experiments an aero piston engine for small air-vehicles—a well-made and reliable powerplant—connected to a propeller. This paper presents the need for a better sensing and measurement system. Then, we show the development of the system, the measurement protocol and process, recording and analysis of the data, and results of some experiments. We then demonstrate the possibilities this sensing suite can achieve—a deeper insight into the operation of the machine—by performing high-quality analyses of engine cycles, well beyond capabilities in the state of the art. This system can be generalised for other rotational machines and equipment.

## 1. Introduction

High-fidelity characterisation of operation of rotary machines (engines, motors, gear boxes, etc.) has been conceptually straightforward but difficult to achieve in practice.

The literature—detailed in Section 3—has demonstrated that a knowledge and technology gap has existed in successfully implementing instant angle speed (IAS) measurement systems for rotational equipment and engines. The existing systems are severely limited either in or a combination of maximum speed, resolution, or quality of measurement.

The development and capability demonstration of the system developed and described in this paper were conducted on a small internal combustion engine (ICE), a very complex and constrained testbed, in order to validate the concept.

Internal combustion engines (ICE) are complex thermo-mechanical systems that have been used for the last century and a half as powerplants in a myriad of applications. Even if they were studied extensively, there are still many areas in which our knowledge about ICE is limited. The bulk of engine development has taken place before computational power and sensing capabilities were available or affordable.

ICE started being replaced with electric motors when the electrical power distribution systems became available. However, there are vehicles, machinery, and equipment where the specific qualities of ICE will still make them the solution of choice for many decades to come, warranting further study and research effort into improving their operation [1].

Better characterisation of ICE operation provides the potential to expand their performance envelope and significantly reduce losses and their side effects such as noise, vibrations, excessive wear, and pollution.

A system for measuring an ICE in its real operating environment represents a challenging problem. It shall be capable of performing, in real-time, the measurement of all relevant variables under load, similar to—or in real—operating conditions. These requirements demand an objective approach for a system, with a dedicated design, which shall produce consistent results that can then be analysed, and new knowledge inferred.

As is shown explicitly in the analysis in Section 2, there is an acute need for better measurements for ICE, as required by legislation relative to the harmful side effects of the engines over the full extent of environmental and operating conditions.

This paper presents an innovative measurement suite—a specialised sensor, the data acquisition system, and the data processing methodology—that will permit new insights in the operation and performance of rotational equipment and ICEs. This is a critical enabling technology for further developments in engine technology and optimisation of operation of rotational machinery. Table 1 denotes the most used technical terms.

The novelty of the proposed method and its implementation is in the granularity and quality of the measurement, which will be shown to be unmatched in terms of the state of the art, achieving around one order of magnitude better results in resolution, quality, or sampling speed of the measurement, showing evidence of the emergence of new features related to mechanism and combustion performance.

The paper is organised as follows: Section 2 provides the background of the problem and the need for better measurements. In Section 3, a thorough literature review demonstrates there is a knowledge and technology gap in measuring IAS for engines and rotary equipment. The experimental setting is presented in Section 4, with signal processing and data analysis detailed in Section 5. The potential to extract meaningful information and new knowledge from the measurements we performed is presented in Section 6, where some engine cycle (EC) states are analysed in detail. Section 7 concludes the paper.

## 2. Analysis of ICE Characteristics and the Need for Better Measurements

The Law of Conservation of Energy, together with Newton’s motion laws, govern the operation of an ICE. The fully automated thermal machine (ICE) generates mechanical power output, resulting from controlled fast combustion. When equipped with a propeller, such as in our experiments, the engine creates propulsion as a reaction force.

Several small ICEs, generally destined for the hobby scene, remote control aircraft, and UAVs, were run extensively in our research. They are small, but very well made, robust, reliable, and give consistent performance. We analysed them over tens of hours of sustained testing at all regimes.

The tests confirmed the limitations in the state of the art when using dynamometers—the gold standard for engine testing—to do measurements at the level of an (EC).

During routine measuring and characterisation of a small, single-cylinder aero piston engine, Saito FG-40 [3] using a high-end Hysteresis Dynamometer Magtrol ED-715-5 [4,5,6], Figure 1, the typical speed and torque characteristics are like the ones in Figure 2.

A single cylinder engine was used in this research to facilitate direct connection between measurements and engine cycles.

Despite the audible pitch of the engine while running being reasonably constant (indicating relative constancy of speed for set periods) until it was changed to another regime, as shown in Figure 3, when examining the output more carefully, we found that further analysis was warranted. The *x*-axis in Figure 3 represents the time dimension.

To extract the most meaning from the dynamometer rig, we varied the speed. We attempted to measure and characterise the performance of the engine in operation per one EC [7].

There is an empirical rule in the literature of once-per-degree-of-arc sampling rate minimum condition that would be generally considered adequate for “engine cycle-resolved” in ICE [8]. This rule typically refers to CFD, fluid dynamics, and combustion analyses, measurements, and validation in the cylinder in operation.

On the basis of this designation [8], we use the definition for a resolved engine cycle, a cycle in which the measurements of the specific characteristics can be sampled at least once per degree of arc, i.e., a minimum 360 measurements per revolution.

It was clear that the high-end dyno we used has a technical limitation in this regard: at the highest sampling resolution, the max frequency for recording was 100 Hz.

Figure 4a shows the quasi-static characteristic of the engine over about 16 s. Between seconds 88 and 95, the speed was relatively constant. The difference between cycles was visible. Higher magnification of the period between 89.5 and 92 s (Figure 4b) shows that the dyno essentially reached its maximal time resolution. The red dots of the characteristic trend represented the recorded values, which were around 10–12 per EC (i.e., per two revolutions of the camshaft).

Critically, the dyno performed correctly for its intended purpose. However, we attempted to extract extra data and infer additional information from the unit. It was clear that this was possible only at a limited level and specific testing regimes.

Figure 5 presents the level of detail at the point where the dyno reached its limits in terms of resolution. Figure 5a depicts the characteristic at about 1250 revolutions per minute (r/m)—the minimum stable—steady rate. Each EC was visible and could be separated from the next one, with the level of self-similarity of cycles being quite clear, although differences between periods were also evident.

Figure 5b shows that at higher engine revs, the dyno could not produce good results on the speed evolution per EC. Above 3000 r/m, the cycles were sampled randomly, insufficient observations per EC were collected, and the detail disappears and was lost entirely for the peak/minimal speed per cycle.

The data sampling speed required was far higher than 100 Hz (dyno limit) to distinguish the shape of each EC and even higher for an engine cycle-resolved (minimally a sample per degree of arc).

This lack of detail at higher speed, e.g., around 3000 r/m vs. consistent results at 1250 r/m for an engine that has a max effective speed of 9000 r/m, demonstrates the need for a better measurement system.

There is a need for better capabilities for faster sensing and faster recording, inherently, a system capable of achieving a much finer granularity in the measurement process. The measured angular velocity with appropriate resolution and accuracy over the entire EC (the engine-resolved cycle), not available hitherto in the literature, could reveal significant events (such as misfires, knock, or possible wear, or the approach of a catastrophic failure) and their effect on engine performance, besides an engine not performing at its best.

The results due to misfire or a series of misfires can reveal a fast evolution and potentially catastrophic event in case of single-cylinder or even multi-cylinder engines powering an airframe or a mission-critical plant or equipment.

The identification of any less-than-ideal operation regime of an engine can trigger pre-set scenarios to prevent/control failure or keep the engine at peak performance. Thus, the development of such sensing suites is critical for mission-critical applications.

## 3. State of the Art in Measuring the Instant Angle Speed

This section summarises a thorough critical review of the state of the art in IAS measurement. Most examples are for ICE, due to the large variation of speed over a cycle and the wealth of information a good IAS can provide.

In measuring IAS and its evolution, various types of measuring instruments and equipment were used: dynamometers, optical sensors, proximity switches, piezoelectric sensors, laser doppler velocimetry, sound/vibration, ion current, in-cylinder pressure measurement, eddy current, IR cameras, etc. Moreover, a measurement system requires adjacent equipment as PC/Laptop for recording the data; various software packages, e.g., Excel, MATLAB or dedicated systems for data/signal conditioning; processing/analysis; lab environmental conditions; consumables; instrumentation; and technical assistance/support. When we refer to new sensors, these include static and dynamic calibrations, validation, and homologation.

The state of the art in the field of measuring and studying the evolution of the speed of engines and other rotational equipment in operation is summarised in Table 2. It has been demonstrated that various research teams did measure engine/equipment cycles, with different levels of success. These levels represent an indicator of the gap of information in having a real-time measured EC.

This literature review was focused on instantaneous rotational speed analysis and control of rotary machinery from a cyclo-non-stationarity perspective [41].

The cycle-resolved attempts reported in the literature focus mainly around the ignition/combustion moment in time relative to the engine shaft angular position. This is where the spike in in-cylinder pressure, torque, and speed occur, relating directly to the combustion energy release. However, this aspect is becoming very clear only when the rest of the cycle is shown, and it is presented later, in Section 5.

The most suitable type of sensor to measure fast angular speed variation proved to be the incremental timer/counter-based optical sensor [13,42] that is attached externally to the engine for non-contact and non-invasive dynamic measurements.

The data sampled this way needs to be analysed further for determining the effects of variation of the angular velocity relative to the engine performance per each EC.

Table 2 highlights the limitations of different systems as reported in the literature. These limitations are a combination of speed, resolution, error level, and consistency of measurements.

In a previous work presented in [7], our research team attempted to use a dedicated Hall effect compound sensor (HECS) to measure the variation of crankshaft angular speed for a series of complete engine cycles. The HECS controlled by an Arduino DUE [43] as the measuring platform consisted of 12 Hall effect sensors [44] placed on a fixed radial clearance and angular pitch location (a_0_, a_1_,…, a_11_) of the propeller mounting bush having a rare earth magnet for triggering/controlling the ignition timing. Moreover, the compound sensor was reading the peak of magnetic field value at a sampling rate of 500 µs and a response time of 3 µs. An Arduino DUE acquired the data and recorded it on an SD card, and it was analysed further in MS Excel.

The characteristics and direct analysis of the raw data demonstrated very clearly that each cycle of the engine was different from any other cycle and that there was a high similarity between ECs. However, the HECS system was severely limited in its capability, with the limitations originating primarily in phenomena at higher frequency measurement (fast variation of the magnetic field) and space, which precluded the setting of more sensors, and consequently a higher resolution of the system.

What was required was a novel method and measuring system to explore the profile between the points and investigate further and in more depth the phenomena inside the engine cycle, which we expected and anticipated from the data available at that point (dynamometer tests and the HECS).

From this analysis of the state of the art, although the technology and components of systems able to perform high quality and high-resolution IAS measurements in engines and other rotational equipment have existed for a few years, it is evident that there is no robust, reliable, high-resolution, and high-quality methodology and implementation for IAS measurement. Much effort has been expended into trying to detect specific patterns in operation that can be discovered from high-quality measurements, recording, and analyses of IAS. Most effort was directed at developing methodologies and software tools to extract meaning from data rather than on robust systems that generate that IAS data.

A 1985 NASA report [45] demonstrated the origins of the complexity of ICE operation and why novel experimentation techniques and computation are needed to achieve an even relatively crude understanding as a step towards successful modelling the ICE physical processes. A 2020 editorial in the *International Journal of Engine Research* [1] highlighted the need for deeper analyses of engine combustion, including transient phenomena such as cycle-to-cycle variations that are not well understood or analysed. This is predicated on better measurements and characterisation of IAS than what is available at this moment.

This paper highlights our implementation that attempts to fill this gap, namely, developing a robust, reliable, high-resolution, and high-quality methodology and implementation for IAS measurement.

## 4. Experimental Setting

### 4.1. Description

The experimental system developed for this research and used for measuring the variation of the speed of the crankshaft per each engine cycle consists of an engine [3] propeller system to be measured as it is performing in real conditions, a codewheel [46] and an optical encoder [47], a signal acquisition system which includes an oscilloscope [48], a test bench, and auxiliary equipment (Figure 6).

The components of the experimental system are represented schematically in Figure 7. 

The test bench secures the engine and permits a safe operation. Auxiliary equipment includes a safe stop button and the engine starter. A portable tachometer was used as an independent instrument for reading and recording the speed when stabilised (once per experiment), as well as to verify, separately, that the results are within reasonable limits.

This experimental system evolved over several iterations. Previous attempts, using an Arduino-based signal collection and recording system, even with several evolutionary generations of hardware and software implementations, proved unsuitable due to the rich information content of the signal (well beyond capability of even the Arduino DUE [43]).

### 4.2. Engine-Propeller System

A small single-cylinder aero piston-engine equipped with a propeller constitutes a simple platform for measuring engine shaft angular velocity variation in operation. The thermodynamic cycle inside the engine drives the propulsion. This simple approach avoids complexity and misinterpretations by eliminating other uncontrolled variables.

The propeller, as a sink of mechanical power, has well-defined characteristics and smooth performance curves. The system engine-propeller produces consistent results, as the angular velocity constitutes an integrator of all processes during the cycle and its interaction with the airstream. It consists of
-A four-stroke engine—naturally aspirated gasoline, conventional spark-ignited internal combustion, small aero-engine—Saito FG-40 [3], cylinder AAC (A—aluminum piston, A—to an aluminium cylinder and C—Hard Chrome plated), static cold compression ratio 3.5 bar (measured after run-in), 3.5 HP (≈2.2 kW measured using a dynamometer). The engine characteristics are as follows, based on the manufacturer’s specifications: Bore Ø 40.0 mm, Stroke 32.0 mm, Displacement 40.2 cm^3^, Practical speed approx. 1700 –9000 r/m, Recommended Propeller 19” × 10”~21” × 8”. The engine runs on fuel made by a volumetric mixture of 20 parts of unleaded petrol ULP 91 to one part fully synthetic engine oil (simplifies engine construction).-A propeller—the engine is equipped with a Classic Series propeller (18 × 8 in), diameter 457 mm (18 in), pitch 203 mm (8 in), hub thickness 22.2 mm (7/8 in), shaft diameter 7.94–9.53 mm (5/16–3/8 in), weight 144.9 g (5.11 oz), the direction of rotation standard/tractor, material: glass fibre R composite.

It needs to be noted that although the engine we use is small, it was built very well and produced consistent and repeatable results over many tests performed at different regimes. Saito FG40 is an engine for small hobby airplanes, it is not a toy engine.

### 4.3. Optical Sensor

The angular velocity sensing system consists of an optical sensor working as an incremental encoder that converts the angular position of the shaft into an analogue signal.

The OE is mounted securely on the engine block and powered by a 5V DC Power Source Ps. We used a HEDS-9100#A00-500 counts per revolution (CPR) unit [47].

The choice of the 500 CPR OE correlates with the capability of the signal processing system. We used only one channel for this research.

The system generates an analogue signal that feeds into the signal acquisition system—to an oscilloscope (Os), being visualised, recorded, and processed further.

The codewheel [46] is mounted securely on the propeller’s bush. Its specifications are as follows. Type HEDS 6140#B09–1000 CPR, metal; Max velocity 500 r/s; Max acceleration 250,000 rad/s^2^, Max count frequency 100 kHz.

The max count frequency of 100 kHz and the 1000 slots/revolution code wheel permits the sensing of speed and speed variation of up to 100 r/s. For measurements at higher speeds, a glass code wheel is recommended (that has a limit of 200 kHz).

The typical output of the combination Cw-OE as used in our experiment is presented in Figure 8. The blue and the red characteristic correspond to the two channels.

It is worth noting that the distancing of the detectors was not in opposition (as it would typically be for a 1000 CPR OE instead). The two sets of sensors k were distanced at π/2 electric angle.

### 4.4. Signal Acquisition System

#### 4.4.1. Introduction

The output data of the sensing system—Cw and OE—need to be collected, recorded, processed, and interpreted. This process was done in the signal acquisition system and then through data processing as illustrated in Figure 9.

The signal from the optical sensor (OE) [46,47] system is fed to an Os [48], which acquires, converts, displays, troubleshoots the measurement, and records the signal on a flash drive for data storage and further analysis.

The instrument of choice for measuring the instantaneous speed at fine granularity was the oscilloscope. Previous attempts to use other acquisitions systems—such as the Arduino DUE—failed due to bandwidth limitations.

We used a Tektronix MSO4054 oscilloscope with an analogue bandwidth of DC to 500 MHz and 10 million samples memory [48]. With the oscilloscope reading time interval of 10^−6^ s, the system collects and records the voltage signal—as presented in Figure 8—over 10,000,000 observations (i.e., a length of an experiment of 10 s).

The Cw, with its 1000 CPR, enables achievement of a fine granularity measurement at a rate of 1000 electric cycles per one revolution or 2000 electric cycles per one EC. This resolution exceeds the accuracy of “the once-per-degree sampling rate (generally considered adequate for cycle-resolved)” [8] by nearly 2.8 times. The reading speed of the signal controlled by the oscilloscope is well within the capabilities of the instrument we use.

The oscilloscope acquires the signal at pre-set frequencies, converts the analogue output signal from the optical encoder into a digital signal, and stores it onto a USB flash drive for further processing.

At this point, the system can measure engine speed fluctuations at high enough resolution while it practically operates at any speed.

#### 4.4.2. Measurement Process

The typical measurement process (with reference also to Figure 9), needs to proceed through the following critical steps:i.The Os needs to be carefully set and calibrated as per manufacturer’s instructions to avoid measurement errors and needs to be isolated from vibrations.ii.The engine is started, warmed up, and is running at a stabilised speed (approx. 2400 r/m for the example presented below).iii.The 5V DC and the Os are both connected with the OE, first to the input, and second to the output A and B ports and the common ground. Because the Os is a highly sensitive instrument, grounding errors are likely to produce significant errors and required careful consideration.iv.Channel A and B ports of the OE linked with the Os at a preset sampling interval—e.g., 10^−6^ (s) or (1 MHz) sampling frequency and a record length of 10,000,000 observations are setting the length of data recording of the oscilloscope, i.e., 10 s in this example.v.The signal is acquired and recorded in a comma-separated variable (*.csv) file and transferred to the USB flash drive for storage.

The table header and the first five samples visualised in MATLAB is presented in Figure 10. It contains three columns, the first records the time, the second—voltage on Channel A, and third—voltage on Channel B, as per line 15—TIME (timestamp), CH1, and CH2 (data for the voltage signal from the two channels of the OS, as the Cw slots pass in front of the photodetectors). First five observations are shown on lines 16 to 20, the rest from six to the end are not shown.

Early attempts at postprocessing the recorded output signal in Microsoft Excel proved that Excel has hard limitations, being difficult to work with more than about 1,000,000 observations. As a result, we opted for MATLAB, where there are no such limitations so that processing could continue. MATLAB also has a multitude of useful signal processing tools and a dedicated signal processing toolbox.

#### 4.4.3. Measurement Results

Figure 11 represents a sample of the voltage signal imported and analysed using MATLAB for a typical engine run, visualised as the signal for one channel.

The horizontal line at 2.5 V is a reference value of the measurement points through the experiment (highlighted red dots) that are superposed on the blue signal line. They represent the connection of successive recorded values of the voltage while the sampling was taken. In this particular case, the red dots are 1 microsecond apart on the abscissa, with a direct interdependence between the observation number and time. Therefore, the measurement points, evenly spaced on the abscissa, have various coordinates on the ordinate proportional with the amount of light that reaches the detector.

This time variation requires an interpolation operation to determine the time difference between the signal crossing the threshold, i.e., the horizontal line of 2.5 V constant reference value, e.g., signal rising as presented in Figure 11.

The up points u_1_, u_2_, to u_i_ are the time coordinates where the signal line crosses the reference line when the recorded signal increases. The down points d_1_, d_2_, to d_i_ are the time coordinates where the signal line crosses the reference line when the recorded signal decreases in voltage. This tandem detailed in Figure 11 permits the computation of the time differences between pulses and the instantaneous speed of the engine shaft.

The signal needs significant processing to produce meaningful and useable results that can permit its use in the analysis, operation, and optimisation of an ICE.

#### 4.4.4. Precision of the Measurement

For the recorded signal, as in Figure 11, on the *x*-axis—observation number linked to time—the Os’s long-term sample rate and delay time accuracy is ±5 ppm over any ≥1 ms time interval [48].

To determine the precision of the signal—the output voltage of the OE proportional with the light modulated by the Cw as read by the Os—we used the output signal over an experiment (10 million data points). This signal can be used to determine the precision of the whole sensing system that includes the OE (with its 5 V DC supply), the Cw (axial and radial runout), Os (dynamic measuring accuracy), probes, connectors, and insulators.

We used the maximum values of the signal for the full aperture of the opening—the voltages read by the Os—as the measured points of interest and as an invariant. The following methodology was used to determine the precision:The input is the ten million points recorded of an experiment.The signal was cut at 4.5 V (all values below 4.5 V were set to zero).For each subset of values corresponding to an output greater than 4.5 V (the tips of the signal), the maximum value was calculated.The set of maxima was then analysed. We determinedthe average for the experiment (for 137236 peaks) was 5.07327 V;the standard deviation was 0.00287 V, with the natural (±3 sigma) variation of the signal being 0.01722 V, which was less than 0.3395% of the signal range.

We conclude that the sensing suite was very high quality and the measurement—which includes all measurement systems, from 5DC supply through to OS, probes and cables, Os, ADC conversions, and output—is very precise.

## 5. Signal Processing and Data Analysis

### 5.1. Introduction

The *pulse-period* function in MATLAB automates the interpolation of the intersection points n_1_, n_2_, n_3_…n_i_ between the falling signal and the reference horizontal dashed line at half of the 5 V OE output, as presented in Figure 12. Their x_1_, x_2_, x_3_…x_i_ coordinates are computed as the pulse-period intervals that permit the computation of the time differences between pulses and, subsequently, of the instantaneous speed of the engine shaft.

The time between each two-consecutive intersection *n*_i_ points between the signal and the reference line is calculated, and then the IAS can be computed.

The following algorithm is used to calculate the speed of the engine:

1. Load data (10,000,000 observations).

2. Establish the reference signal line (we used 2.5 V as the reference value).

3. Determine pulse-period *S*_i_ (Table 1) of the counts difference between every point *n_i_* and precedent one *n_i−_*_1_, where the falling signal line intersects the reference line and record the delta values (Figure 12).

4. Compute the time difference between each pair points *n*_i_–*n*_i−1_ successively within a resolution of 0.000001 s.

5. The result is used to compute the instantaneous frequency (IF) of the IAS variation between each window/bar pair using the formula:(1)$$n_{i}=\frac{C_{w}}{S_{i}\cdot s_{f}}\;(\mathrm{Hz})$$
where *n*_i_—is the number *n* of revolutions per second (Hz) [2];

*C_w_*—is the constant codewheel angular pitch;
(2)$$C_{w}=\frac{1}{1000}\left(r\right)=0.001\left(r\right)=\frac{2\pi }{1000}\left(rad\right)=0.00328318\;\left(rad\right)=\frac{360}{1000}\left({}^{\circ}\right)=0.36{}^{\circ},$$

*S*_k_—pulse width IAS modulated Electric Cycle Span (Table 1);
*S_k_* = x*n*_i−1_ − x*n*_i−2_ (counts),(3)

*s*_f_ = 0.000001 (s)—Os sampling frequency rate;

First pulse-period is
*S*_1_ = x*n_2_* − x*n_1_* = 19801.8891 − 19779.4813 = 22.4078 (counts)

Therefore, the first total time/pulse or per one period/electric cycle *S*_1_ is
*t*_1_ = 22.4078 · 0.000001 = 0.0000224078 (s).

The second is
*t*_2_ = 22.6023 · 0.000001 = 0.0000226023 (s),
the third is
*t*_3_ = 22.2585 · 0.000001 = 0.0000222585 (s),
and so on to the last cycle t_2000_ (not shown).

The S_k_ presented in Figure 12, by definition, has 360 °e (electrical degrees) with three interpolation points defined by the first cross point *n*_i−2_ = 0 °e, the middle point n_k_, and the third cross point *n*_i−1_ = 360 °e.

Considering the case of a slow rotation with *n* near to 0 (Hz), the signal would rise on a vertical line at 180 °e at *p*_k =_ *S*_k_/2. In this case, the signal tilting angle α_k_ shows the offset of the middle point n_k_ by Δ*p*_k_. This indicates the presence of a speed at a rotation *n_k_* with α_k_ in a range less than 40 °e, which is the maximum signal tilting angle (p. 4 *Encoding Characteristics* Case 1 Max. [47]).

The total period (*t*_ci_) for one EC_i_ (two revolutions of the crankshaft) represents the sum of the 2000-time intervals using the formula:
(4)$$t_{\mathrm{ci}}=\sum_{i=0}^{2000}S_{i}\cdot s_{f}\;(s),\;$$
where *t*_ci_ is the total period for one EC_i_, and *S*_i_ (Table 1) is the period between two successive *n*_i_ points (in Figure 12). Moreover, one complete *t*_ci_ evolves over two revolutions, each having 1000 CPR with 360 °e per one CPR, therefore over 720,000 °e per 4π (rad).

The IAS variation *n*_i_ (Hz), can be also used to calculate the instantaneous angular velocity (IAV) ω_i_ (rad/s), or the instantaneous angular acceleration (IAA) *n*_ai_ (r/s^2^).

For each experiment, an independent and separate tachometer speed check was performed for verification purpose.

### 5.2. Engine Cycles

Plotting the instantaneous speed for the whole experiment of 10 million data points generated the characteristic shown in Figure 13 for 199 complete ECs from this data sample.

Moreover, the average signal output of approx. 40 kHz (equates to 40 Hz × 1000 CPR) presented in Figure 13 shows that the reading frequency in this case is well below half of recommended sensor’s maximum operating frequency of 100 kHz and well before derating curves are recommended for correcting the results [47]. The sensor was installed and operated as per manufacturer’s instructions.

To note: we can increase the resolution of measurements by increasing the sample rate for the current setting, but this would come at the expense of proportionally reducing the number of ECs recorded. Thus, we would reduce the capacity to see the big picture and a reasonable number of ECs, as well as to infer more general conclusions.

Figure 14 represents a detail of 12 successive ECs, 178 to 190 from Figure 13, with their angular velocity variation.

The engine characteristic operation depicted in Figure 14 incorporates a wealth of information that needs to be analysed in detail, and that can lead to some very interesting findings.

### 5.3. Results and Discussion

The IAS—sensing suite, the measurement system, and the data processing—presented in this paper permit unmatched insight into the workings of rotational equipment and ICE. This subsection presents a synopsis of some of our observations thus far for the case of an ICE, with more inferences to be conjectured and publicised as the research develops.

The observations below are mainly qualitative due to the substantial amount of modelling of dynamics and structures to reach the quantitative appreciation of the forces that cause shaft velocity variations. They will be detailed in future publications.

The thermodynamic cycle in the ICE’s combustion chamber drives the piston, connecting rod, crankshaft, and the power consumer—here, the propeller—at an angular velocity with a typical variation shown for one EC in Figure 15 (i.e., for 2000 samples—two engine shaft revolutions marked with vertical grid lines from 0 to 4π). On this graph, one can identify the differences between the theoretical and real-time dynamically measured points. The characteristic represented in Figure 15 is that of a complete EC—one observation to every 1/1000 revolutions. This resolution largely exceeds the requirement of 1 observation per degree to qualify as a cycle resolved.

The *x*-axis represents the observation number (#). The *y*-axis represents the angular speed of the engine. The scale for *y*-axis is adjusted to permit better visualisation of results.

It is essential to note the small irregularities in the signal. We hypothesise that they may originate from three possible sources (in direct relation with the extreme resolution and sensitivity of the sensing and the signal processing system):-small window/bar size tolerances and radial runout;-small vibrations between the optical sensor and the code wheel;-torsional vibrations of the piston–pin–conrod–crankshaft–propeller assembly.

We chose to present the raw data here (that can be smoothed over, see below), with further research regarding the source of non-uniformity in progress. Cleaning the data may destroy the wealth of useful information. The apparent noise in measurement may be a door to much more subtle phenomena that just need extra attention and further work.

Regarding Figure 15, there is a noticeable (and expected) piston acceleration, increasing between top dead centre (TDC) toward bottom dead centre (BDC), i.e., between 0 to π from 37.75 after spark ignition occurs to 41.34 r/s before BDC.

In ideal conditions, in the absence of gas dynamics considerations and the opening and closing of valves, a crank mechanism connected to a flywheel that simply rotates due to inertia has a non-linear characteristic due to acceleration/deceleration of the piston, pin, and conrod.

After ignition, around the point 0, the engine shaft accelerated and reached a maximum speed just before π, then dropped to 40.94 r/s after the exhaust valve opened, increased a little to 41.18 r/s as the combustion gases were released, then decreased to 40.53 r/s after the exhaust valve closes (elasticity and inertia of gases in the cylinder). It built up slightly to 40.73 r/s just after 2π as the exhaust valve closed, and then it dropped to 39.12 r/s after the intake valve opened and the air-fuel mixture flowed into the cylinder due to natural aspiration (negative pressure). It was raised to 39.31 r/s, due to the inrush of air-fuel mixture inertia; then, the intake valve closed just after 3π.

Then, the piston decelerated to 37.65 r/s just before the next spark ignition and due to compression, at 4π; then, the next EC started.

The speed drop of 0.1 r/s between the start and finish of this particular cycle represents engine deceleration. The inflection points a and b represent a relatively steady state. They can be characterised as inflection points in the function’s graph of thermodynamic states, i.e., between generating vs. consuming mechanical energy during the EC, cumulated with rotational inertial energy of moving parts.

The force generated by the inertial mass of the moving parts—crankshaft, conrod, piston bolt and piston assembly inside the cylinder, together with the propeller with its airflow reaction—is characterised by a continuous variation, which, as shown here, can be precisely measured, studied, and monitored.

The detail in Figure 15 represents one increment speed variation in time (acceleration) with a sample increment of 0.000001 (s) out of 2000 observations per EC.

The *x*-axis in Figure 15 represents the observation number, as highlighted above. However, critically for this type of experiment, due to speed variation, the duration of each observation is variable in time. There is no direct correlation on that graph between the number of observations and time.

To make it possible for different ECs to be directly compared with one another, we can transform the graphs from the number of observations as an independent variable to time, correcting the time with the value of speed.

Figure 16 represents the same characteristic of the engine, with corrected *x*-axis to represent the actual time for each observation.

In this case, there are 2000 points on the *x*-axis (full EC, two revolutions), but the points in time for each observation are not equally spaced and are adjusted for speed.

After executing this operation, we obtain a time-correct graph that represents the complete EC time-resolved characteristic, i.e., there are 1000 observation points in time per each revolution and at least one measurement per degree of arc.

The following observation is important: if the subsequent cycle c180 is superposed on the graph in Figure 15, only their speed vs. observation varies, considering 0 as the start point and 2000 as the end point, which are identical on the *x*-axis. At the same time, in Figure 16, in the time domain, higher velocity will correspond to points being moved to the left (higher speed means shorter time between observations). Lower velocities will move characteristics to the right (longer duration between consecutive observations). Fundamentally, for the time speed characteristic, it is the equivalent of time contracting at higher speeds or dilating at lower rates when compared to observation number/speed characteristic.

The data from our observations can be used for much more than what exists in the state of the art (misfire, torque reconstruction). We can in fact determine rotation events associated with the contribution of each combustion cycle to the engine operation when a steady state is achieved or when extraneous influences (from the mechanism or driven plant) occur.

Specific evolutions in the operation of an engine can be determined from the observation of time-speed engine characteristics. Some examples are presented in the following section.

## 6. Engine Cycle States

### 6.1. Acceleration State

Figure 17 represents cycles 189 and 190 from Figure 14 in the same time frame for comparison, for consistency over 2000 observations.

The acceleration can be detected by comparing—visually or computationally—two consecutive ECs. It can be seen clearly on the graph that the duration of the two ECs is different:Δ*t* = *t*c189 − *t*c190 = 0.0488 − 0.0479983 = 0.0008217 (s) 

It is apparent that tc_190_ < tc_189_. Because the angle swept during the EC time is equal, the average angular velocity per EC, ω_c_, is different: n_190_ > n_189_. Therefore, we can calculate angular velocity variation in time as a difference in speed between two firing ECs 189 and 190. We can also calculate the angular acceleration as the average value per EC, as presented in Table 3.

Visually, when examining Figure 17, we can notice that EC 190 had consistently higher speed, over the cycle, than EC 189, and it was shifted to the left.

### 6.2. Deceleration State

Similarly, deceleration can be visualised and calculated as for acceleration. Figure 18 presents the case when the engine decelerated from an EC to the next due to misfire. In this case, *t*_c3_ > *t*_c2_, and speed *n*_3_ < *n*_2_.
Δ*t* = *t*c3 − *t*c2 = 0.0480098 − 0.0472227 = 0.0007871 (s)

Visual examination of Figure 18 demonstrates that, even if the two ECs had a common start point on the *x*-axis, c3 was longer than c2. Moreover, even if the angular velocity of the EC in c2 and c3 were almost superposed at the start on the *y*-axis (angular speed), cycle c3 ended up at 39.44 (r/s) or (Hz), compared to 40.23 (Hz). Moreover, examining the two curves, we found that the cycle c3 curve was mostly below the cycle c2 curve, demonstrating lower speed throughout the cycle. The significant points on c2 were displaced to the left, i.e., occurred earlier in the cycle than the same points on c3.

### 6.3. Steady State

Figure 19 shows a case when the engine operated steadily, and the subsequent fire cycle performed nearly the same as the previous one, with minimal differences.

This state represents the goal for engine control required for efficient engine operation, with minimal additional stresses.

However, this is challenging to achieve without exploiting the instantaneous speed data, as developed in our research and presented in this paper, due to the many uncontrolled and fast-evolving variables.

In this case
Δ*t* = *t*_c179_ − *t*_c180_ = 0.0504204 − 0.0505031 = −0.0000827 (s)
Δ*n* = Δ*n*_180_ − Δ*n*_179_ = 0.1429 − (−0.2925) = 0.4354 (Hz)
Δ*n*_180_ = 37.4975 − 37.3546 = 0.1429 (Hz)
Δ*n*_179_ = 37.3546 − 37.6471 = −0.2925 (Hz)

### 6.4. Misfires

It can be easily noted that in Figure 20 at the zero start, looking at the evolution of the angular speed, the engine accelerated from 39.38 r/s to 44.31 r/s for a normal ignition fire start, followed by small decelerations/accelerations to the end of cycle to 40.13 r/s, in centre of the graph. It was followed by a misfire, and what should be an ignition followed by combustion at the start of the new EC was just a bounce-back to 42.44 r/s, due to the energy from compressed gas inside the cylinder vs. the inertial mass of the moving parts, i.e., engine-propeller. The misfire cycle led in a drop of angular speed from 40.13 r/s to 38.57 r/s for the cycle.

This drop represented a reduction of 3.9% of the speed, but a drop of 7.6% when measuring the kinetic energy of the system (proportional with the square of the speed of moving parts). This drop in kinetic energy is significant for mission-critical engines, as a series of a few misfires can fatally compromise the system, leading to an engine that stops and cannot restart, lacking the energy to compress the mixture and create work.

It can also be observed that a normal fire cycle c6 is shorter than the following cycle c7, which shows a misfire. Also, cycle c7 has a lower instantaneous speed and, consequently, takes longer to execute the whole revolution.

### 6.5. Unicity of Each Cycle

Figure 21 is a representation of all superposed 199 complete cycles shown in Figure 13, starting from 0 reference point for each EC. Each curve in the figure is an individual EC but smoothed for easier observation.

Because the beginning of the record was random, and the start of the experiment was marked as 0 in Figure 13, even if the full record was for 200 complete cycles starting with the intake stroke, only 199 cycles were complete ECs starting with the expansion stroke. They all started with the lowest speed point onto the codewheel and they could be extracted from the record and presented at the same starting point in Figure 21.

This figure demonstrates that each complete EC of the engine for the length of this experiment (199 ECs) was unique in duration and amplitude. The misfiring sequences were also visible—as the flatter, slower ECs, compared to the longer ones.

Each EC has a unique signature, but high commonality with other ECs, permitting relatively simple mathematical modelling (this is work in progress, to be published later). It is also clear that there is a reasonable EC period commonality. The length of EC as measured between the shortest and longest EC were similar, which is expected, considering that the speed of the engine is relatively constant in our experiment.

## 7. Conclusions and Future Work

The sensing suite and the methodology developed in our research and presented in this paper can measure, with very high resolution, the instantaneous angular speed of a rotational machinery operating at high speed.

The experimental implementation was done using a small piston aero engine working in realistic conditions, incorporating the most complex combinations of constraints for a rotational machine we could find: high and continuously variable speed, high speed gradients, heavy requirements on resolution and precision, and high level of vibrations.

The high speed, high-resolution, high-quality measurements were conducted over a cycle and over multiple cycles at a granularity level that permits an unmatched level of insight into the operation of ICE. The system allows differentiation of performance for each cycle and parts of a cycle.

This possibility of fine granularity measurement has demonstrated the capability to collect 1000 speed records per revolution, with 2000 samples per EC record. Our research was able to achieve full engine cycle measurement of the internal combustion engine. Most methodologies and approaches developed for IAS analysis or using IAS in the state of the art and presented in Section 3 can be applied to analyse the data generated in our experiments.

With reference to Table 2, we demonstrated the methodology at a speed of about 2400 r/m, and a sampling resolution of 1000 samples/r, with the natural variation (±3 sigma) of less than 0.3395% of the signal peaks range, correlated with a tachometer measurement and with consistent and repeatable results. Therefore, we achieved results at least an order of magnitude better than the state-of-the-art in terms of resolution, quality, or sampling speed of the measurement.

By adjusting the combination code wheel, sampling frequency of the oscilloscope, and the oscilloscope memory, we can greatly expand this methodology up to 500 r/s and 200 kHz count frequency of the optical encoder.

This type of measurement system sets the foundation for a new research-level capability that can establish details on engine power production/density per cycle. Quantifying this information would allow researchers to reach a new level of understanding of the dynamic performance of the engine in real time, controlling and optimising its operation.

A few observations can be summarised from the data collected and processed in this paper for application of the proposed system to an ICE:-The angular velocity of the engine has a continuous variation over a cycle, between a minimum, just before the point when ignition takes place; with a steep acceleration, the speed reaches the peak and then slows down until the next ignition event.-Each cycle is unique—over many experiments we ran, we were never able to observe or record two identical cycles. This is consistent with the combination and distribution of values of the multiple variables that affect the operation of an engine, as highlighted in the 1985 NASA report [45].-Even if each cycle is unique, there is a high similarity between cycles, potentially facilitating data interpretation and use for engine control.-The information collected permits characterisation of each cycle, opening the possibility for more in-depth analysis of the operation of the engine.

Future work will involve the following:-High-resolution quantitative and comparative analysis of the data collected, which, due to extent of the paper, needs to be published in a follow-up article.-Automation and speeding-up of data collection, with the aim of achieving real-time data collection, processing, and control.-Identification of trivial vs. critical variations of speed that would require prompt intervention.-Finer, more detailed analysis of the data per cycle and comparisons between cycles.-Mathematical modelling of the engine operation with identified and controlled variables that will improve the match between model and actual output per cycle.-Identification of unusual patterns in operation and strategy to correct them.-Development of high-speed control loops capable of stepping in and correcting a potential failure as it develops.-Monitoring the evolution of the engine as it wears and the cycles change.-Optimisation of the engine operation to reduce emissions, noise, vibrations, shocks, fatigue, and wear.-Extension of the measurement system and methodology to multi-cylinder engines of various architectures and engine cycles.-Extension and generalisation of the measurement system and methodology to any rotational machine and equipment.

The measurement capability presented in this paper is a key enabling technology that opens the way to control each revolution or cycle separately, in order to achieve desired uniformity or evolution of speed (accelerations and decelerations) with minimal/nil adverse effects.

## Figures and Tables

**Figure 1 sensors-22-00754-f001:**
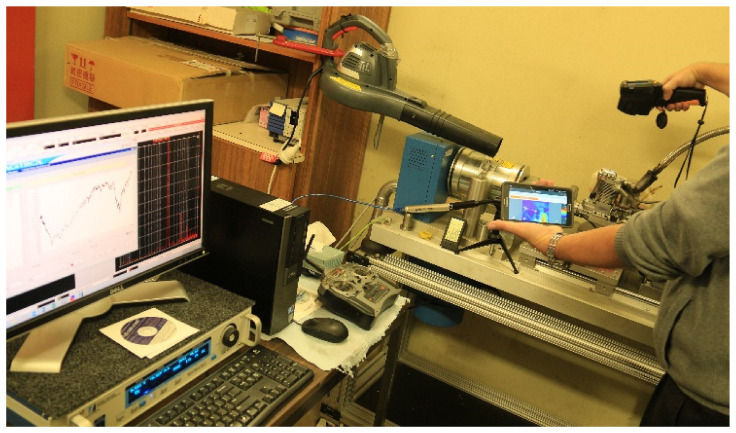
Dynamometer experiments on a Saito FG-40 engine.

**Figure 2 sensors-22-00754-f002:**
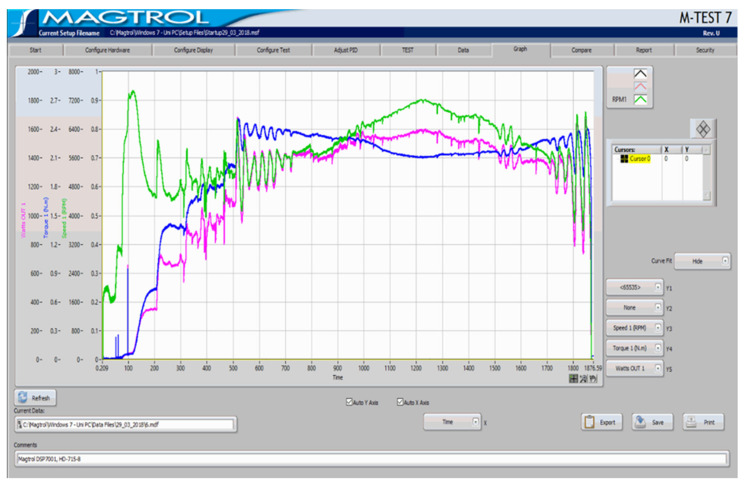
Evolution of the Saito FG-40 engine during an experimental dynamometer run.

**Figure 3 sensors-22-00754-f003:**
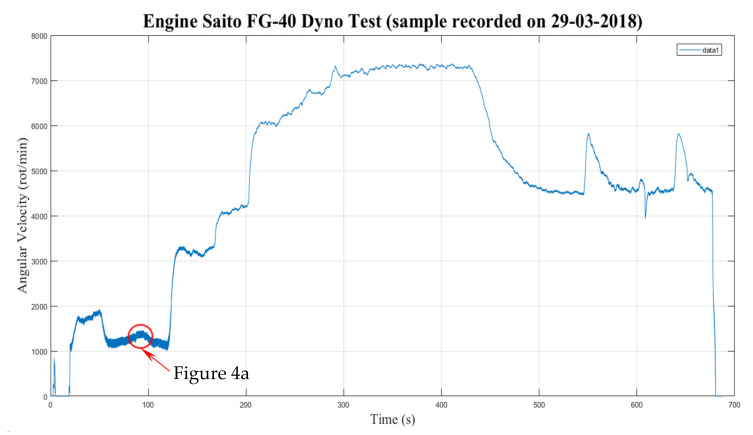
Speed evolution of the Saito FG-40 engine during an experimental run.

**Figure 4 sensors-22-00754-f004:**
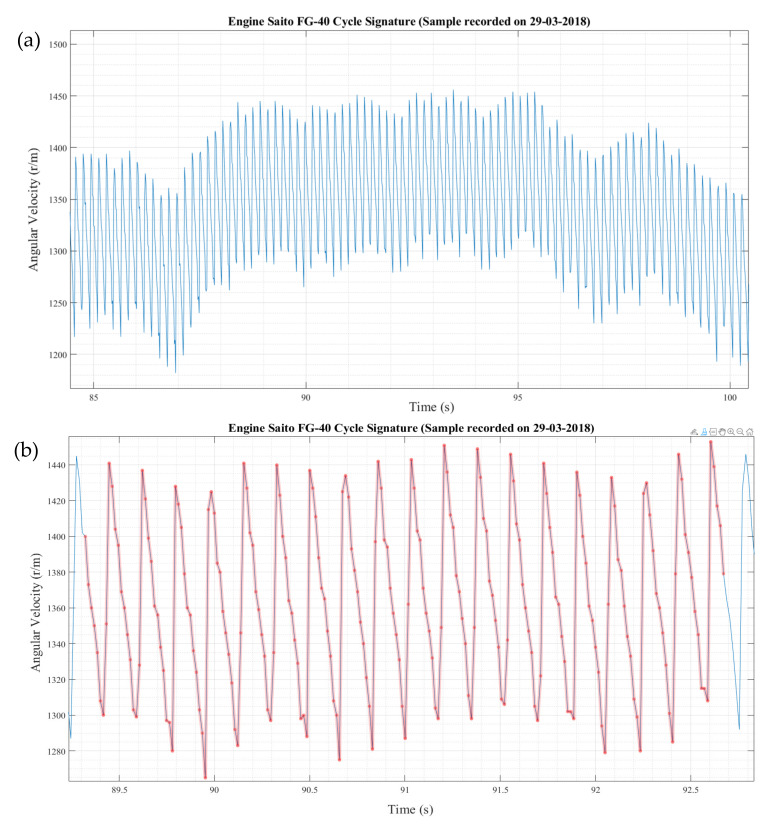
Fine details of speed evolution of the Saito FG-40 engine.

**Figure 5 sensors-22-00754-f005:**
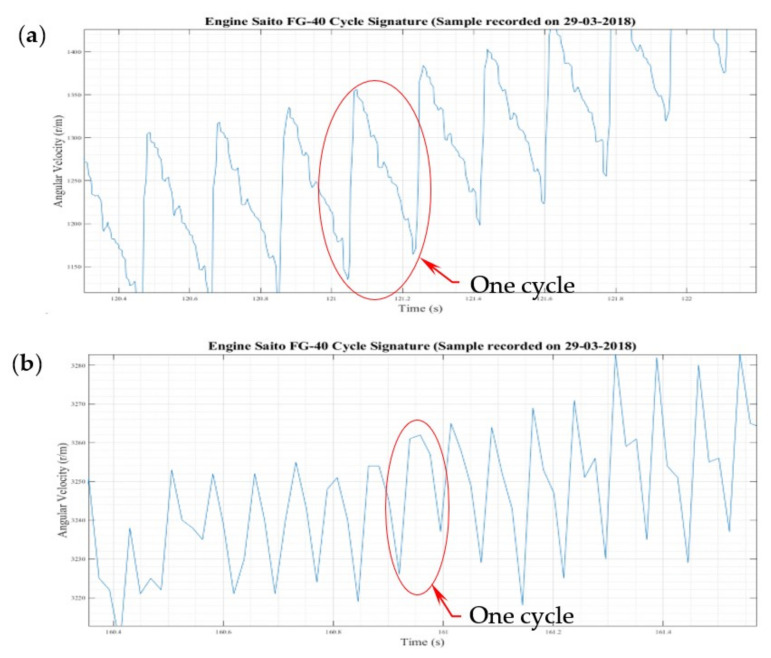
Fine details of speed evolution of the Saito FG-40 engine (**a**)—1250 r/m; (**b**)—3240 r/m.

**Figure 6 sensors-22-00754-f006:**
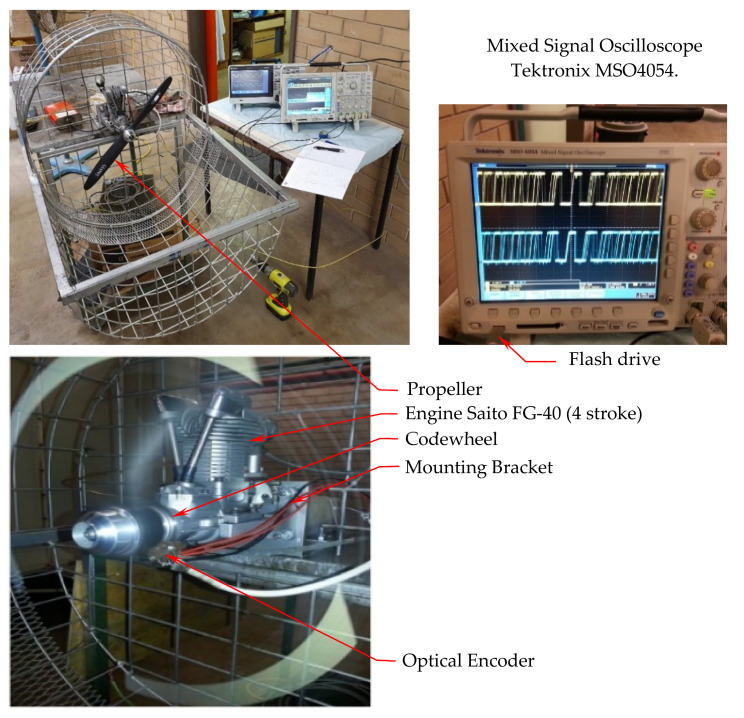
Engine test bench.

**Figure 7 sensors-22-00754-f007:**
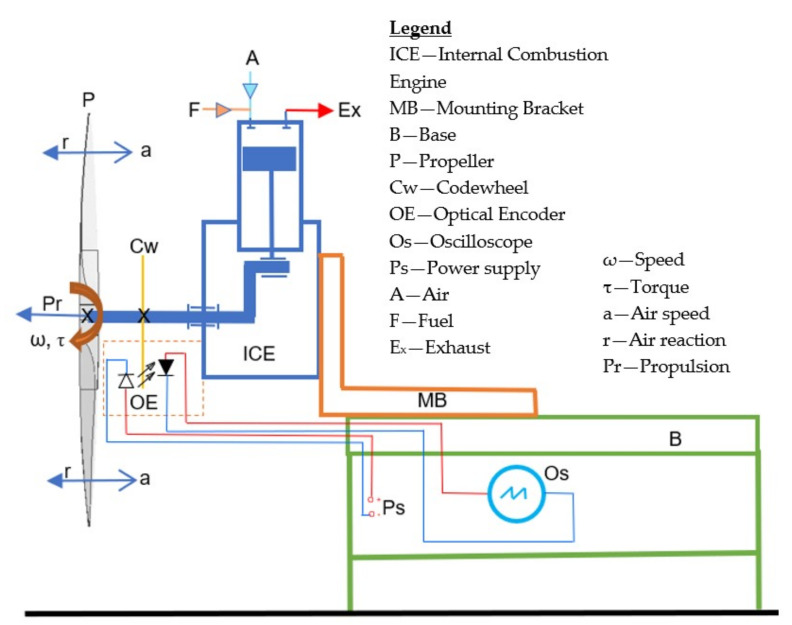
Experimental setup schematic diagram.

**Figure 8 sensors-22-00754-f008:**
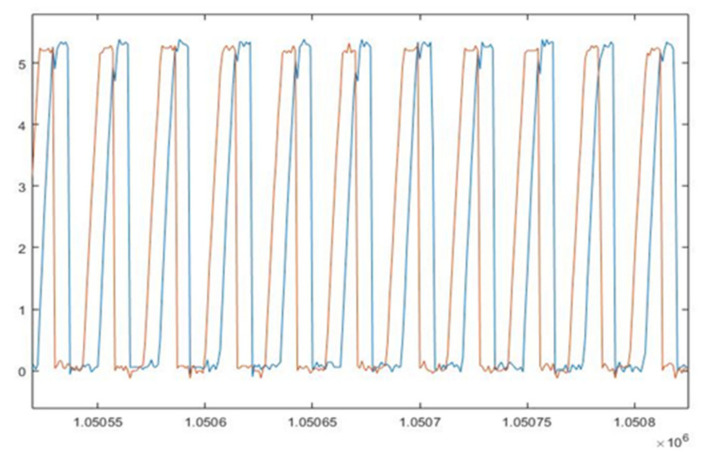
Optical sensor output channel: A—red, B—blue.

**Figure 9 sensors-22-00754-f009:**
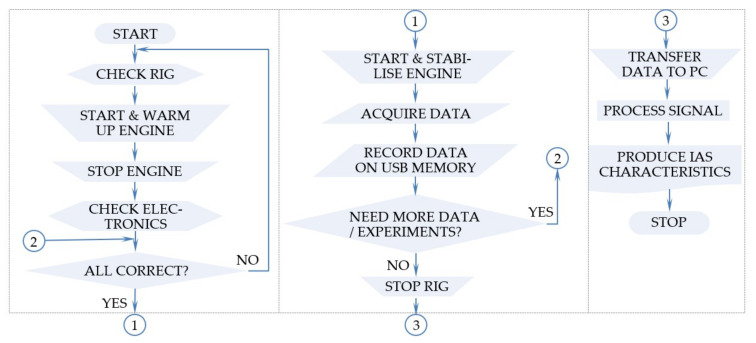
ICE operation IAS signal flowchart.

**Figure 10 sensors-22-00754-f010:**
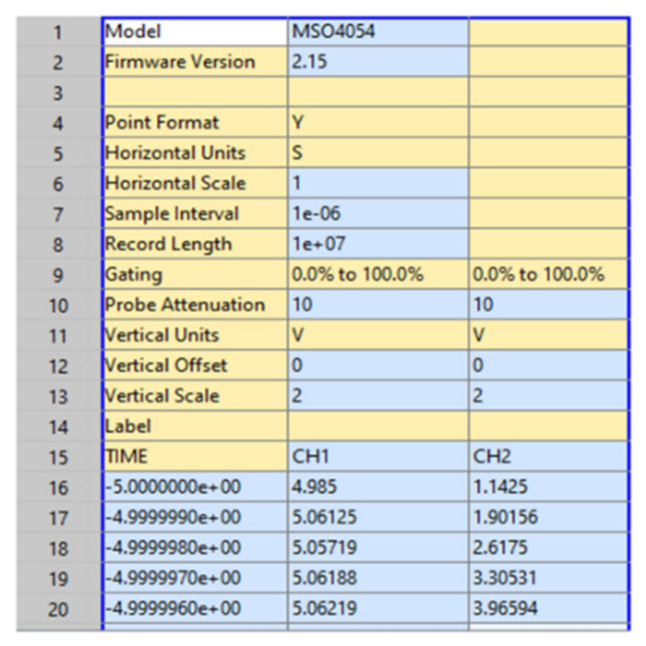
Measurement data—extract.

**Figure 11 sensors-22-00754-f011:**
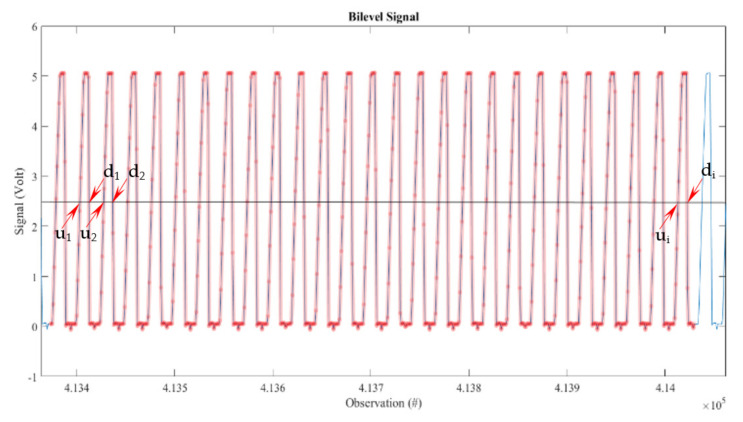
Standard encoder voltage output vs. observation.

**Figure 12 sensors-22-00754-f012:**
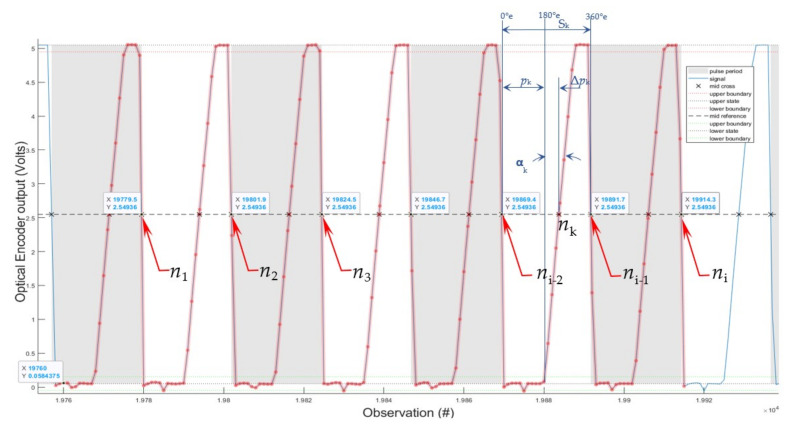
Signal interpolation.

**Figure 13 sensors-22-00754-f013:**
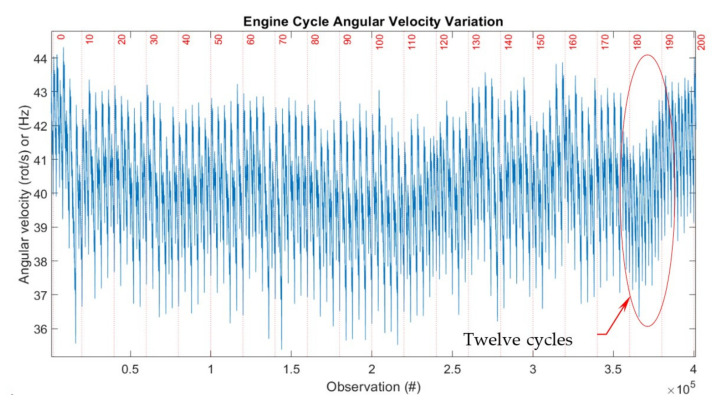
Representation of 199 consecutive engine cycles. # stays for the number i.e., Observation number.

**Figure 14 sensors-22-00754-f014:**
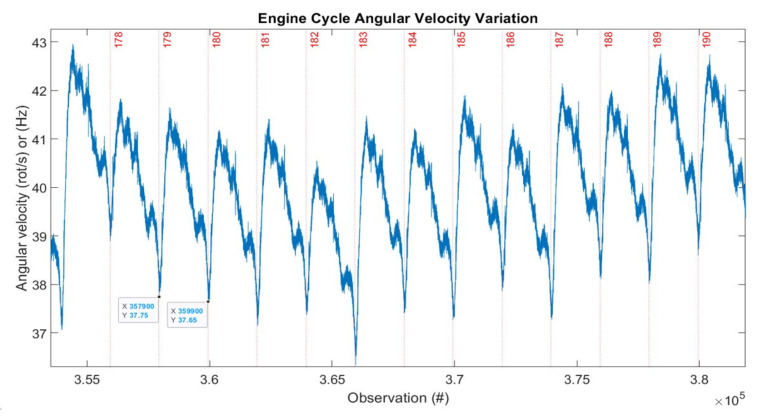
Representation of 12 consecutive engine cycles c178 to c190. # = number.

**Figure 15 sensors-22-00754-f015:**
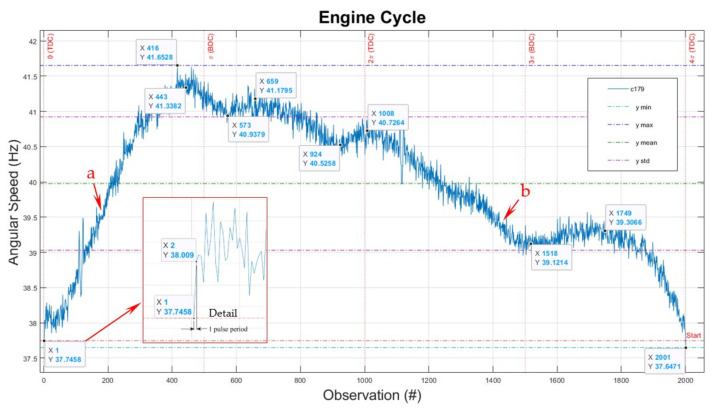
Engine typical cycle signature—observation numbers. #—number.

**Figure 16 sensors-22-00754-f016:**
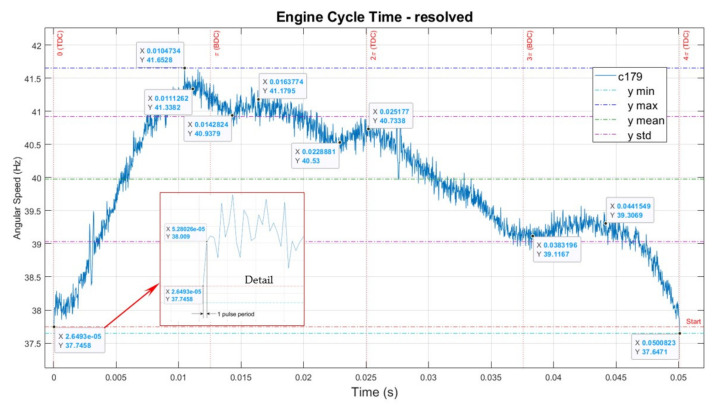
Engine typical cycle signature—time scale.

**Figure 17 sensors-22-00754-f017:**
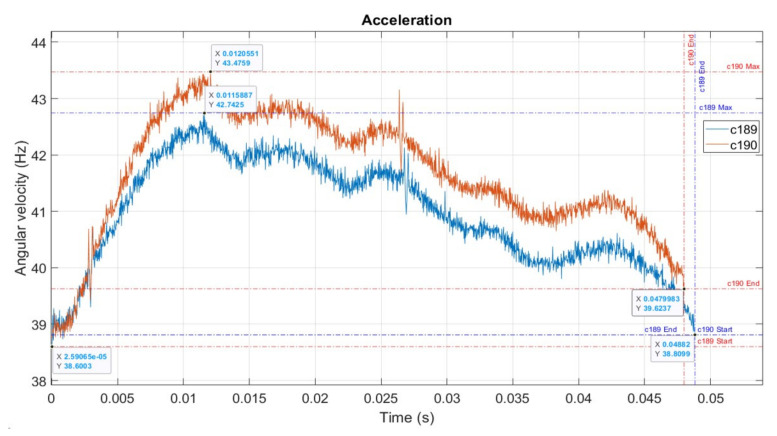
Cycles 189 and 190.

**Figure 18 sensors-22-00754-f018:**
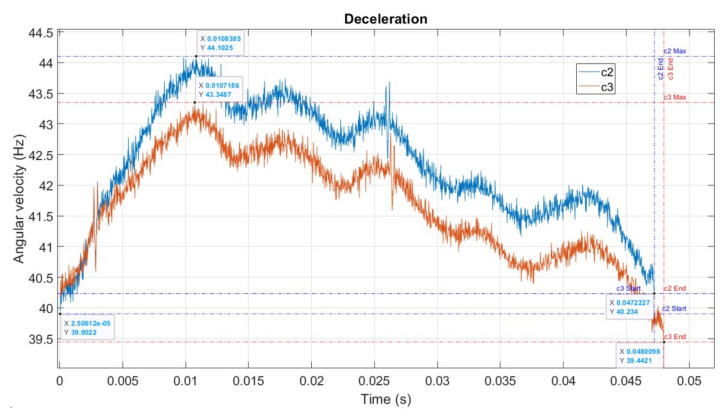
Cycles 2 and 3.

**Figure 19 sensors-22-00754-f019:**
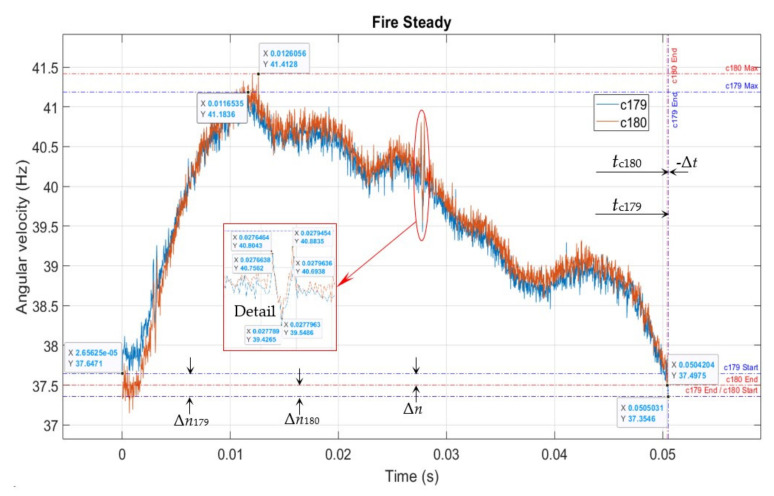
Cycles 179 and 180.

**Figure 20 sensors-22-00754-f020:**
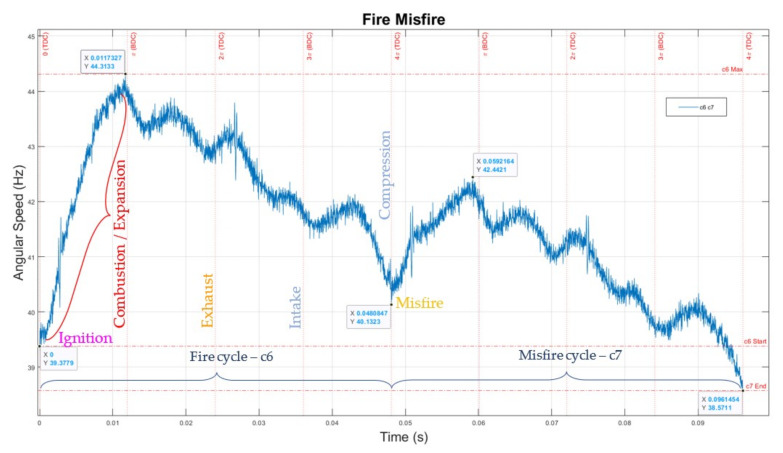
The engine in operation: c6—fire, and c7—misfire cycles.

**Figure 21 sensors-22-00754-f021:**
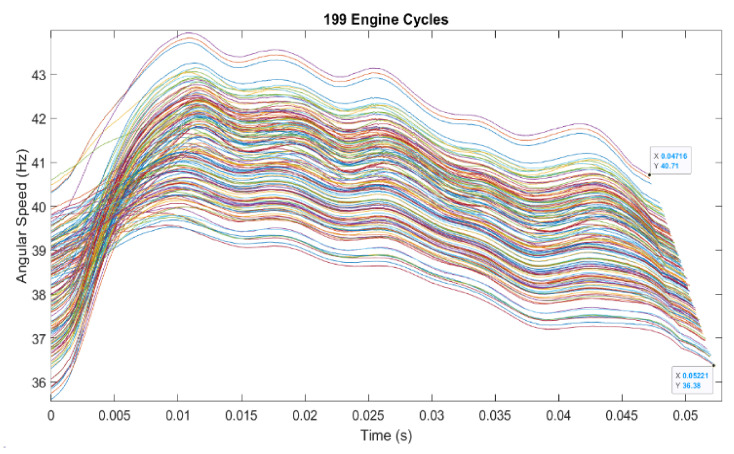
The engine in operation—199 visibly unique cycles.

**Table 1 sensors-22-00754-t001:** Abbreviations.

Abbreviations and Acronyms	Meaning
ADC	Analog-to-digital converter
BDC	Bottom dead centre
CFD	Computational fluid dynamics
CPR	Counts per revolution
S	Electric cycle span
Cw	Codewheel
EC	Engine cycle
HECS	Hall effect compound sensor
IAA	Instantaneous angular acceleration
IAS	Instant angle speed [2]
IAV	Instantaneous angular velocity [2]
ICE	Internal combustion engine
IF	Instantaneous frequency
OE	Optical encoder
Os	Oscilloscope
TDC	Top dead centre
UAV	Un-crewed aerial vehicle

**Table 2 sensors-22-00754-t002:** Instantaneous angular speed measurement methodologies.

Sensor Type [Ref.]	Measurement			Precision		Correlation with Other Methods, i.e., Dyno	Produce Consistent Repeatable Results
	Input r/min	Sampling Resolution/r	Output Signal	Error Analysis	Error Level σ (%)		
OE [9]	200–1000	2048	V	Y	2	-	-
OD [10]	900	60	V	Y	24	Y	-
SM [11]	1044	50	Hz	Y	-	N	-
SS [12]	0–3000	60	V	Y	1.7	Y	Y
SC [13]	750	5fe	ADC	Y	0.1	N	N
OE [14]	1500	1000	V	N	N	N	Y
OE [15]	600–1800	360	V	N	N	N	Y
ME [16]	5400	64	V	N	55	Y	Y
ZT [17]	0–4500	63; 57; 104	V	Y	±0.2	Y	Y
MPS [18]	1500–4500	ANN	Graphs	Y	10	N	Y
OS [19]	1000–1800	180	V	Y	-	N	-
MP		122					
OE [20]	750	400	TTL 300 kHz	N	-	N	Y
OE [21]	125	1024	V	N	N	N	Y
OE [22]	500	720	V	N	N	N	Y
– [23]	1180	-	-	-	-	N	Y
EM [24]	800–1400	122	-	N	N	Y	-
HS [25]	1050	-	-	N	N	-	-
HS [26]	800; 1500	125	V	Y	10	N	-
HS [27]	2990	120	V	Y	-	Y	Y
SE [28]	2200	1000	bar	-	-	Y	Y
AE [29]	2000–5000	360	-	Y	2–14	Y	Y
MP [30]	1250; 3250	59	V	Y	55	Y	N
OE [31]	1556	360	V	Y	1.485	N	Y
OE [32]	1556	360	V	Y	1.485	N	Y
OE [33]	60	1000	V	Y	7	Y	Y
OE [34]	750	180	V	Y	10.5	N	Y
OE [35]	800–2000	720	V	N	N	Y	-
VS [36]	30–50 Hz	25.6 kHz	Hz	Y	0.6	N	Y
PA [37]	-	100 mV/g	V	Y	100	Y	Y
PP [38]	3000	120	V	-	-	Y	Y
AVS [39]	993	N	N	Y	3	N	Y
SS [40]	650–850	N	-	N	N	N	-

**Where:** optical detector—**OD**; stepper motor—**SM**; speed sensor—**SS**; soft counter—**SC**; analog to digital converter—**ADC**; optical encoder—**OE**; magnetic encoder—**ME**; zebra tape—**ZT**; magnetic pulse sensor—**MPS**; artificial neural network—**ANN**; optical switch—**OS**; magnetic pickup—**MP**; magnetic inductive transducer—**MIT**; electromagnetic sensor—**EM**; hall-effect sensor—**HS**; shaft encoder—**SH**; angle encoder—**AE**; vibration sensor—**VS**; piezoelectric accelerometer—**PA**; proximity probe—**PP**; angular velocity sensor—**AVS**; yes—**Y**; no—**N**; not indicated **–**; voltage—**V**; frequency of occurring event—**fe**.

**Table 3 sensors-22-00754-t003:** Acceleration calculation for ECs 189 and 190.

EC	189	190	Delta
*t* _c (s)_	0.04882	0.047998	8.217 × 10^−4^
*n* _avg (r/s)_	40.96682	41.66814	7.013 × 10^−2^
*ω* _avg (rad/s)_	257.402	261.8086	4.40656
*n_a_* _(__r/s_^2^_)_	853.50558		
$\dot{\omega }$ _(_ _rad/s_ ^2^ _)_	5362.73372		

where *t*_c_—EC duration, *n* _avg_—number *n* of revolutions per second (average), ω _avg_—angular velocity (average), *n_a_*—number *n* of revolutions per second squared (average), $\dot{\omega }$—angular acceleration (average).

## Data Availability

Not applicable.

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
