# Peer review of "Robust Instant Angle Speed Measurement for Internal Combustion Engines—A Novel Sensing Suite and Methodology"

_sensors, 2022, doi:10.3390/s22030754_

Round 1

Reviewer 1 Report

The reviewer’s comments on the paper are as follows:

Major revisions

  1. More than 40% of the paper is an introduction and literature review. It does not seem to be a scientific article, but a beginning of a monograph. This part of the paper requires aggregation. No reference is needed to figures from other sources, this confuses the reader (e.g. Lines 337 and 338). Arrange the references in the order in which they appear in the text.
  2. Can the described measuring system be applied to larger engines?

Minor revisions

  1. Page 2, Table 1. Why is the acronym “Ck” for “Electric Cycle” not as logical as the others? Make a reference to the table in the text.
  2. Page 5, Fig. 4. Describe the graphs axes legibly, extend the caption under the figure (a, b).
  3. Page 7, Lines 177 to 181. Change the table number to 2. What does “*” mean for [30] in the first column?
  4. Pages 8 to, Lines 195 to. Standardise the style of writing the names of the authors of the cited papers.
  5. Page 14. Change table numbers.
  6. Page 18, Line 588. ni.
  7. Page 23 and 24. Change table numbers.

Author Response

Dear Reviewer,

Please find below our response relative to your comments provided:

Major revisions

  1. More than 40% of the paper is an introduction and literature review. It does not seem to be a scientific article, but a beginning of a monograph. This part of the paper requires aggregation. No reference is needed to figures from other sources, this confuses the reader (e.g. Lines 337 and 338). Arrange the references in the order in which they appear in the text.

Response

The state of the art’s extensive description was reduced, keeping Table 2 and adding a summarising sentence (lines 190-192). The extended literature review was a response to some previous reviewers’ requirements, but we fully agree it was a bit much.  

Reference figures removed from the text Lines 337 and 338.

References are rearranged in sequential order.

  1. Can the described measuring system be applied to larger engines?

The system was developed and demonstared for a single cylinder to facilitate a direct relation between measurements and engine cycle – lines 93 – 94.

It  can be applied to larger engines and to any rotational equipment – this is part of the future work – lines 735-738.

Minor revisions

  1. Page 2, Table 1. Why is the acronym “Ck” for “Electric Cycle” not as logical as the others? Make a reference to the table in the text. (Corrected, thank you.)

Response

The acronym was changed with “S” from the Electric Cycle Span in Table 1 Page 2, in Fig. 12 and the afferent text lines on pages 15 and 16 in order to avoid confusion with the Engine Cycle (EC); thank you.

  1. Page 5, Fig. 4. Describe the graphs axes legibly, extend the caption under the figure (a, b).

Response

 Rectified, thank you.

  1. Page 7, Lines 177 to 181. Change the table number to 2. What does “*” mean for [30] in the first column?

Response

Table number changed to 2 on Page 7, Line 177, and the “*” that marked the work referred to the Wiebe combustion model was deleted, thank you.

  1. Pages 8 to, Lines 195 to. Standardise the style of writing the names of the authors of the cited papers.

Response

Corrected, thank you.

  1. Page 14. Change table numbers.

Response

Corrected, thank you.

  1. Page 18, Line 588. ni.

Response

Page 15, Line 415 ni corrected thank you.

  1. Page 23 and 24. Change table numbers.

Response

Corrected, thank you.

Thank you so much for being so helpful. We have revised the paper, and we appreciate and acknowledge your help – line 753.

Reviewer 2 Report

This manuscript is devoted to the novel sensing approach to robust instant angle speed measurement for internal combustion engines. This manuscript is well organized, and the topic is actual. The authors presented an essential analysis of the characteristics of internal combustion engines and justified the need for measurement improvements. The state-of-the-art is presented in the third section and provides an exhaustive overview of modern research in the selected area. The authors proposed a signal acquisition system and conducted experiments on signal processing and data analysis. The examples are good, and the research is undoubtedly worth the attention. However, I recommend adding several things to clarify the research.

  1. A flow chart with the procedures of the proposed methodology should help us understand the whole technique.
  2. The difference between the present study and previous works in the selected area should be highlighted.
  3. Limitations and highlights of the proposed methodology should be clearly described in the Discussion.

Despite the issues mentioned above, I think this manuscript is worth considering for acceptance.

Author Response

Dear Reviewer,

Please find below our response relative to your comments provided:

This manuscript is devoted to the novel sensing approach to robust instant angle speed measurement for internal combustion engines. This manuscript is well organized, and the topic is actual. The authors presented an essential analysis of the characteristics of internal combustion engines and justified the need for measurement improvements. The state-of-the-art is presented in the third section and provides an exhaustive overview of modern research in the selected area. The authors proposed a signal acquisition system and conducted experiments on signal processing and data analysis. The examples are good, and the research is undoubtedly worth the attention. However, I recommend adding several things to clarify the research.

  1. A flow chart with the procedures of the proposed methodology should help us understand the whole technique.

Response

A flow chart at Line 311 was added – Thank you.

  1. The difference between the present study and previous works in the selected area should be highlighted.

Response

Table 2 Page 7 shows onto the last three columns the precision, correlation with other methods and if the previous works produce consistent, repetitive results compared to the present study. Also, the Discussion starting at Line 182 highlights the findings and limitations of each work. Thank you.

  1. Limitations and highlights of the proposed methodology should be clearly described in the Discussion.

Response

The present work was resubmitted following editor advice based on the input from three reviewers. Therefore, it reached an extensive 28 pages. We hope that the Results and discussions at Line 489 completed by the Conclusions and Future Work in section 7, Line 675 highlight the limitations and future work required to cover the new research ground in full.

Thank you so much for being so helpful. We have revised the paper, and we appreciate and acknowledge your help – line 753.

Round 2

Reviewer 1 Report

No further comments.

This manuscript is a resubmission of an earlier submission. The following is a list of the peer review reports and author responses from that submission.

Round 1

Reviewer 1 Report

The paper presents the use of an encoder to measure instantaneous engine speed. This is obviously nothing new. Also the observations referring to the relationship between instantaneous speed and the combustion process are trivial. The paper is too long and the discussion on the sensor choice is useless, as optical encoders are pretty standard in every R&D test cell.

The discussion on the insights gained on the combustion process based on instantaneous speed observations is superficial, and 30 years of literature on the subject (misfire detection, torque reconstruction, etc.) are already available.

I think the paper should not be published.

Author Response

Dear Reviewer

Thank you for reviewing our work and advising the need to emphasize the key point of difference between existing sensor implementations and what we did here.

We would like to add the following observations and indicate the changes we have made: 

The measurement system is destined for rotational machinery in general, with the use of an IC engine to illustrate implementation. We have clarified this with the following text (in red below)

This paper presents an innovative measurement suite – a specialized sensor, the data acquisition system, and the data processing - that will permit new insights in the operation and performance of Internal Combustion Engines that is a critical enabling technology for further developments in engine technology and optimization of operation of rotational machinery. (lines 61-62 in the paper)

The novelty is in the granularity and quality of the measurement, which is unmatched in the state of the art. We have access to a high-end dyno and the capability of the described system is at least an order of magnitude better. A thorough and up-to-date literature search (one reason why the paper is extensive) demonstrates that there is a gap in measurement and characterization capability which is where our solution sits. We have made it clear that the contribution is the order of magnitude increase in sampling speed and the emergence of new features related to the mechanism and combustion performance. We have added the text (in red below)

The novelty of the proposed method and its implementation is in the granularity and quality of the measurement, which will be shown to be unmatched in the state of the art, achieving around one order of magnitude better results in resolution, quality, or sampling speed of the measurement and we see evidence of the emergence of new features related to the mechanism and combustion performance. (lines 63-67 in the paper)

The observations we made about the speed and influences on it - combustion being but one -  are mainly qualitative at this stage as the paper is already long. But our observations are well confirmed by theory and the theory demonstrates many models use a broad brush – correlation factors – to compensate for the lack of measurement and characterization capability (An Introduction to Thermodynamic Cycle Simulations for Internal Combustion Engines by Jerald A. Caton, Wiley, 2016, Chapter 13) – in which coefficients are used to compensate for the lack of data. The more extensive and deeper insight we have will be published in our future papers when the data about combustion and friction can be captured in a controlled study.

We have added the text below:

The observations below are mainly qualitative due to the substantial amount of modelling of dynamics and structures to reach the quantitative appreciation of the forces that cause shaft velocity variations. They will be detailed in a future publication. (lines 691-693 in the paper)

The data from our observations can be used for much more than what exists in the state of the art (misfire, torque reconstruction). We can in fact determine rotation events associated with the contribution of each combustion cycle to the engine operation, when a steady state is achieved or when extraneous influences (from the mechanism or driven plant) occur. (lines 769 – 773 in the paper)

We express the hope that we clarified some of the points raised in the review. We appreciate the observations made were useful for improving the clarity of the paper.

Thank you! 

Reviewer 2 Report

Very well structured manuscript focused on fine granularity measurement. Congrats to the authors.

State of the art is relating the nowadays solution and the research interest area, also the references are the majority of the past 10 years which reveals the up to date research.

Good experimental support for the sensing suite and methodology proposed.

The work is based on a very well done expert interpretation of the signal data acquisition.

The fine granularity measurement capability can be useful for future researchers in order to reduce time in detecting faults.

The paper is interesting and can be published in the journal as it is.

Author Response

Dear Reviewer,

We are grateful for the review and comments.

Thank you! 

Reviewer 3 Report

The paper is interesting. The authors provide a novel technique to measure the the Instantaneous Angular Speed of an engine over a cycle and over multiple cycles. Upon the extracted experimental results, the methodology was found stable and showing well predictions. However, some points should be addressed to improve the submitted article. Several suggestions are listed as follows:

  1. The introduction is very brief. I suggest the information in Section 3 to remove and to add in Introduction section.
  2. In measurement technique: Fired engine  tests are highly problematic
    in terms of accuracy. They suffer from cycle-to-cycle variations as well as of noise issues and are often limited to low engine loads. Also, the
    employed filtering is always an intentional modification of the results. Filtering is always a degree of freedom to fit the results to the desired output and is, therefore, a major weakness. The authors should be determined the information of sensors and the relative errors during experiments.
  3. I strongly urge that it is necessary to more strongly mention major shortcomings regarding technique in both the extracted results and in the conclusion.

Author Response

Dear Reviewer,

 Thank you for reviewing our work.

 Here are the answers to the issues highlighted: 

  1. We accept that the introduction is brief, but we wanted to just set the scene and refer the reader to various specialized sections. We think that adding Section 3 to the Introduction would extend it excessively. We are directing the reader to Section 3 in lines 31-35 and the last paragraph of Section 1. We emphasized this in (lines 31-35 and 72 in the paper).

The literature – detailed extensively in Section 3 – has demonstrated that a knowledge and technology gap existed in successfully implementing Instant Angle Speed (IAS) measurement systems for engines and rotational equipment. The existing systems are severely limited either in or a combination of maximum speed, resolution, or quality of measurement.    (lines 31-33 in the paper).

In Section 3 a thorough literature review demonstrates there are a knowledge and technology gap in measuring IAS for engines and rotary equipment. (line 73 in the paper).

  1. The data we obtained and provided/displayed in the paper is not filtered in any way. The only place where the data was further processed in Fig 23, where the engine characteristic was smoothed to detect the identity of each cycle. The errors were determined relative to an invariant – the full aperture of the opening.  We have added this in Lines 621-623

              We used the maximum values of the signal for the full aperture of the opening - the voltages read by the Os - as the measured points of interest and as an invariant. (lines 604-605 in the paper).

  1. The limits of our paper are implicitly presented as future work in Section 7. We added the following:

 - High resolution quantitative and comparative analysis of the data collected, which, due to extent of the paper, needs to be published in a follow-up article (lines 899-900 in the paper)

Thank you!

Round 2

Reviewer 1 Report

My evaluation was 'reject' and I don't think a revision would change the following considerations:
- the subject is not of scientific relevance: the authors propose something (a crankshaft encoder) that is in use in every test bench since I graduated (unfortunately, more than 20 years ago...);
- the observations based on instantaneous speed are superficial: in the late 90s and early 2000 many papers focused on the subject of extracting information on instantaneous torque and possibly cylinder pressure from engine speed. The authors roughly explain that there exist a correlation between torque and speed, without using equations;
- the section dedicated to the implementation (using Arduino, etc.) should not be placed in a scientific paper: it is a part of 'behind the scenes' work that must be done to get there, but results trivial in a paper

Reviewer 3 Report

The revised paper is now acceptable.